# Immune Response to Herpes Simplex Virus Infection and Vaccine Development

**DOI:** 10.3390/vaccines8020302

**Published:** 2020-06-12

**Authors:** Anthony C. Ike, Chisom J. Onu, Chukwuebuka M. Ononugbo, Eleazar E. Reward, Sophia O. Muo

**Affiliations:** 1Department of Microbiology, Faculty of Biological Sciences, University of Nigeria, Nsukka, Enugu 410001, Nigeria; muosophia0@gmail.com; 2Department of Biological Sciences, College of Liberal Arts & Sciences, Wayne State University, Detroit, MI 48202, USA; chisomjoshua982@yahoo.com (C.J.O.); reward.eleazar@wayne.edu (E.E.R.); 3Department of Biotechnology, Graduate School of Engineering, Osaka University, Suita City, Osaka 565-0871, Japan; ononugbocm@gmail.com

**Keywords:** herpes simplex virus, immune system, immune response, vaccine

## Abstract

Herpes simplex virus (HSV) infections are among the most common viral infections and usually last for a lifetime. The virus can potentially be controlled with vaccines since humans are the only known host. However, despite the development and trial of many vaccines, this has not yet been possible. This is normally attributed to the high latency potential of the virus. Numerous immune cells, particularly the natural killer cells and interferon gamma and pathways that are used by the body to fight HSV infections have been identified. On the other hand, the virus has developed different mechanisms, including using different microRNAs to inhibit apoptosis and autophagy to avoid clearance and aid latency induction. Both traditional and new methods of vaccine development, including the use of live attenuated vaccines, replication incompetent vaccines, subunit vaccines and recombinant DNA vaccines are now being employed to develop an effective vaccine against the virus. We conclude that this review has contributed to a better understanding of the interplay between the immune system and the virus, which is necessary for the development of an effective vaccine against HSV.

## 1. Herpes Simplex Virus and the Immune System

### 1.1. Introduction

Herpes simplex virus belongs to the *Herpesviridae* family, a group of spherical viruses measuring 120–200 nm. There are two types of Herpes simplex viruses, Herpes simplex virus type 1 and type 2 (HSV-1 and HSV-2). These viruses cause lifelong infections that are among the most common viral infections worldwide [1,2,3]. As part of the global effort to control the infections caused by these viruses, many vaccines have been developed [4,5,6,7,8]; however, to date, none has been licensed for use in humans. Following the successful development and use of vaccine against varicella zoster virus, a closely related virus in the same viral family, there has been a recent upsurge of interest in developing vaccine against HSV. It is believed that one of the major problems with vaccine development against HSV is the complex interactions that exist between the immune response and the virus. The immune system which consists of an innate and adaptive component has cytosolic sensors which sense the DNA of this virus and consequently stimulate inflammatory response against it. Intriguingly, HSV possesses a repository of arsenals that ensures its successful replication in the human host. The interplay between these arsenals and the immune system determines an outcome. In this review, we explore how the knowledge of the immune response can and has been used in the development of a functional vaccine against HSV.

### 1.2. Overview of the Immune System

The human body is equipped with an immune system that acts as guard against invading pathogens, which are ubiquitous in the environment. Broadly, the human immune system is of two types, the innate and the adaptive immunity. The innate immunity consists of both the structural component, and the proteins which recognize molecular patterns not present in human cell. They constitute the first line of defence against pathogens. The adaptive immunity is a more specific defence against specific pathogens mediated by the B-cells and the T- cells. One attribute of this immunity is that its effect is long lasting.

#### 1.2.1. The Innate Immune System

The innate immune system is the first point of defence in eukaryotic organisms; it is usually fast and non-specific. It is broadly divided into two, namely, the structural component (anatomical barrier) and chemical component. The structural component involves the skin and the mucus membrane. The skin provides an outer impermeable cover against invasion by pathogens [9]. The skin also secretes chemicals (sweat) and fatty acids which are toxic to invading pathogens and exhibit antimicrobial property. The desiccation and desquamation nature of the skin are also known to prevent bacterial colonization [10]. The mucus membrane is less impermeable compared to the skin. Infection via the mucus membrane involves colonization and ability to overcome the defence of the membrane. The different mucus membranes protect the body against infections using different mechanisms, the upward flapping movement of the cilia in the respiratory tract, the mobility and low pH of the stomach in the gastrointestinal tract and the constant flushing of the urinary tract.

The chemicals components include, the lysozyme, defensins, interleukin, interferon, and complement proteins. Lysozyme is a 1, 4-β-N-acetylmuramidase enzyme present in body secretions such as tears and saliva and mainly acts on bacterial cells [11]. The complement proteins act in a cascade-dependent manner to eliminate pathogens [9]. They can also help in the phagocytic process by opsonizing the pathogen, which facilitates easy uptake by the phagocytes. Interferons are antiviral proteins made by viral-infected cells alongside the lymphocytes. They help establish antiviral state in neighbouring cells thus limiting the dissemination of the viral agent. The neutrophils remove pathogens via production of reactive oxygen species that are toxic to them. They have also been implicated in tumour necrotic factor and interleukin-12 cytokines [12]. Macrophages aid in phagocytising process. Macrophages, alongside dendritic cells, link the innate and the adaptive immunity by processing and presenting antigens to the adaptive immune cells. Eosinophils contain granules that have toxic enzymes and molecules very active against helminths and parasites. Furthermore, pathogen pattern receptors such as the toll-like receptors (TLR) and nucleotide-like oligomerization domain like (NOD-like) receptors play vital roles in innate immunity. They act as sensors of pathogen-associated molecular patterns and alert the cell of an invading danger.

Together, all these work in consortium to prevent the human cells from infection caused by pathogens by removing them and helping to recruit the adaptive immune response.

#### 1.2.2. The Adaptive Immune System

Although the innate immune system is first to attack an invading pathogen, it can be overpowered or tricked by the invading pathogen. At this stage, adaptive immunity is called up to assist. Two distinct features of the adaptive immune system are specificity and memory. Specificity here connotes that it act only against the pathogens that elicited its response, and it keeps a memory of the pathogen for future reference. The first infection takes time for response to be mounted, while in repeated infections, the immune response is faster. The B-lymphocyte and the T-lymphocyte are the principal players of the adaptive immunity. The B-cells produce antibodies and mediates the humoral immune response while the T-cells oversee the cellular immune response.

### 1.3. The Interplay of Herpes Simplex Infection and the Immune System

The host–pathogen interaction is dynamic that has the potential to result in a diseased condition. Factors such as the virulent factors of the pathogen (encoded proteins) could help it in navigating the immune resistance of the host. Toll-like receptors 2 and 3 have been reported to play a role in sensing HSV-1 and the activation of interferon type 1 and inflammatory cytokines [13]. HSV-1 tegument kinase US3 has been reported to inhibit TLR-3 expression and dampen interferon type 1 synthesis [14]. Furthermore, myD88 (a critical adapter molecule of the TLR pathway) levels in the cells are drastically reduced by the infected cell polypeptide 0 (ICP0) of HSV-1 [15]. The ICP0 protein carries out this reduction via its proteosomal and E3 ligase activities. ICP0 is a promiscuous transactivator of HSV-1 immediate–early, early and late genes [16]. ICP0 is an E3 ubiquitin ligase comprising 775 amino acids with an N-terminal RING structure [17]. ICP0 promotes viral replication and reactivation from latency by inhibiting interferon synthesis [16]. This function is dependent upon the degradation of ND10 constituent proteins (PML and SP100) using its ubiquitin ligase activity [15,17,18]. The role of ICP0 protein in HSV-1 infection has been extensively reviewed [19]. The ICP0 protein mutant HSV-1 has been reported to have a low growth rate. Undegraded PML protein stimulates interferon production [17], and this can serve as a platform for an attenuated HSV-1 vaccine.

Additionally, HSV has been reported to inhibit tumour necrotic factor (TNF) alpha NF-kB activation of genes involved in inflammatory response [20]. TNF-α is a cytokine vital in innate immunity and, upon synthesis, induces the expression of genes involved in the inflammatory response. TNF-α binds its receptor TNF-R1 recruiting the adapter protein TNF receptor death domain (TRADD), which then recruits TNFR-associated factor 2 (TRAF2) and receptor-interacting protein 1 (RIP1) to the complex. TRAF 2 modulation of K63-based poly-ubiquitination of RIP1 brings about TGF-β-activated kinase-1 (TAK1), turning on the kinase activity of IκB kinase (IKK) which phosphorylates and degrades IκBα, subsequently leading to the activation of the nuclear factor kappa B (NF-κB), a transcriptional factor [21]. However, HSV-1 _γ1_34.5 protein (late gene encoded) represses NF-κB activation in CD8+ dendritic cells [22]. Furthermore, a tegument protein, VHS, has also been reported to inhibit the viral replication independent NF-κB activation [23]. A HSV-1 protein, UL42, that enhances the processivity of the DNA polymerase, inhibits the TNF-α dependent NF-κB activation [20]. It was found out that UL42 binds the p65 and p50 subunits of NF-κB and inhibits their translocation into the nucleus. This will repress the transcription of genes involved in inflammatory reactions. These can be illustrated diagrammatically, as shown in Figure 1.

Other cytokines such as type I interferon can be activated by the RLR cytosolic signalling pathway. This is produced in response to the IRF3 activation and NF-κB in the RLR pathway. US11 protein prevents RIG-1 and IPS-1 interaction (dimerization of IPS-1 and MDA5) [24]. Similarly, the TLR2/TIR/MyD88/Mal signalling pathway stimulates the NK-Kb transcription factor, which facilitates the synthesis of pro-inflammatory cytokines such as interleukins 6, 8, and 12 [25]. Signalling through TLR2/TIR/MyD88 activates IRF3 and IRF7, which promote the production of interferon-alpha and beta [26].

Furthermore, CD8+ TRM, a subtype of memory lymphocyte which resides in non-lymphoid tissues [27], has been reported to trigger adaptive immune response during HSV-1 infection [28] through IFN-Y and granzyme B effectors. CD4+ cells have been reported to inhibit HSV-1 infection and clear HSV-1 from genital infection sites following primary infection. Naïve CD4+ can differentiate into Th1, Th2, Th17, and induced regulatory T (iTreg) upon interaction with the major histocompatibility complex (MHC) (an antigen complex). The subpopulation that naïve CD4+ T cell differentiates into depends on the cytokine within the native CD4+ environment [29]. CD4+ CD25+ has reportedly been involved in HSV-1 response [30], and the deletion of CD4+ cells increased susceptibility to HSV-1 infection in mice [31]. Yu and colleagues reported that regulatory T (Treg) level positively correlates with viral infectivity and is a requirement in establishing latency [32]. The role of adaptive immune cells in HSV-1 infection has been extensively reviewed in [33].

### 1.4. DNA Sensors as Activators of Host Antiviral Response

Pathogen-associated molecular patterns (PAMPs) are a variety of molecules found on pathogens which are usually recognized by the pathogen recognition receptors and subsequent activation of interferon and chemokines, independent of the TLR and retinoic acid-inducible gene 1 (RIG-1) [13]. Some of the DNA cytoplasmic sensors that have been reported to mediate HSV-DNA recognition include-cyclic GMP-AMP (GAMP) synthase (cGAS)—this is a cytosolic DNA sensor of viral DNA with capacity to recruit the stimulator of interferon genes (STING) adapter protein. The cGAS interacts with DNA via its N-domain, and this interaction is modulated by post-translational modifications such as sumoylation, acetylation, phosphorylation and glutamylation [34,35,36,37]. This interaction causes the synthesis of the chemical messenger cGAMP which binds the endoplasmic reticulum-anchored STING, causing its oligomerization [38]. The oligomerized STING moves to the trans-Golgi, where K27 and K63-linked poly-ubiquitin moieties are added, catalysed by E3 ubiquitin ligase [39]. The K-27 and K-63 call upon the TBK1, which is a kinase that will phosphorylate STING. STING can also recruit TRAF6 to activate the transcription factor NF-κB for expression of genes involved in inflammatory response [40]. Following NF-κB activation, STING gets degraded in the lysosome terminating the DNA sensing. STING activates the interferon regulatory factor (IRF3) and NF-κB signalling pathway to exert its effects [40]. The same authors reported that STING also activates the Jun N-terminal protein kinase/stress-activated protein kinase (JUN/SAPK) pathway.

According to Zhang and colleagues [41] DDX41 (DEAD-box helicase 41), a member of the DEAD-like helicases superfamily (DEXDc) senses DNA in myeloid dendritic cells, which stimulates type I interferon production. It has been reported that the knockdown of the DDX41 gene inhibited mitrogen-activated protein kinase TBK1 and NF-κB transcription factor by DNA, and it depends on STING to sense pathogenic DNA. HSV-1 capsid ubiquitination and Vp5 degradation occur in a proteasome-dependent manner [42]. This liberates the DNA into the cytosol where they get sensed by cytosolic DNA sensors, subsequently inducing the innate immune response. A PYRIN protein interferon gamma inducible protein 16 (IFI16), which functions as a cytosolic DNA sensor that directly binds DNA, has been reported. The study found that the knockdown of the IFI16 gene in mouse models inhibited the activation of transcription factors IRF3 and NF-κB on infection by HSV-1 [43]. This showed that IFI16 is essential for the activation of IRF3 and NF-κB. IFI16 interacts directly with interferon-β, inducing DNA to recruit IRF3 and NF-κB for the transcription of genes involved in inflammatory response. In a recent publication, HSV-1 was shown to also activate the inflammatory response via the nucleotide-binding domain and leucine-rich repeat-containing receptor 3 (NLRP3) [44].

## 2. HSV Immune Evasion Mechanisms

It is critical to consider the mechanisms by which HSV evades the immune system during the process of designing and administration of different vaccine compounds and formats [45]. In fact, the major impediment in the search of a potent cure or vaccine for HSV infection is the myriad of mechanisms through which the virus evades the immune system. Thus, continuous elucidation of these mechanisms is critical to achieve potent prophylactic or therapeutic intervention for HSV.

### 2.1. Modulation of Autophagy

Autophagy has housekeeping roles in regulating normal physiological cellular processes. Degradation of misfolded proteins, cellular differentiation, and defence against pathogens are several of the functions of autophagy [46]. Autophagy can promote innate and adaptive immunity. One study showed that HSV-1 induced autophagy in both immature and mature dendritic cells which are crucial for the induction of antiviral immune responses [47]. The virus selectively targets lamin A/C, B1, and B2 for degradation, which then facilitates the egress of newly formed infectious particles. The same authors also found out that the overexpressing of two proteins KIF1B and KIF2A, which are both members of the kinesin-3 family [48], attenuated this rate of egress via reduced lamin degradation. It has been shown that HSV-1 not only inhibits autophagy in neuronal cells, but also in non-neuronal cells. Interestingly, in the non-neuronal cells, inhibition of autophagy was achieved by the phosphorylation of two autophagy regulators: ULK1 and Beclin1, which are dependent on the HSV-1 protein Us3 ser/thr kinase. The authors also showed that depleting both ULK1 and Beclin1 rescued the replication of the Us3-deficient virus strain [49]. This is significant, as Us3 also plays a role in viral manipulation of cellular apoptosis [50]. Another novel autophagy inhibition mechanism employed by HSV is the downregulation of the mitophagy [51], adaptor optineurin (OPTIN) and the autophagy adaptor protein sequestosome (p62/SQSTM1) [52]. This downregulation is mediated through the E3 ubiquitin ligase activity of the intensely studied HSV-1 immediate early protein ICP0 [53]. It has also been demonstrated that exogenous p62/SQSTMI decreased the HSV viral yield [52], thus confirming a potent antiviral activity, but the exact mechanisms by which the proteins mount antiviral responses still needs to be elucidated. HSV-1 also inhibits autophagy through the protein Us11 [54]. Recently, Liu et al. [55] showed in their study that Us11 disrupts the *tripartite motif* protein 23 (TRIM23)–TANK-binding kinase 1 (TBK1) complex, thus suppressing the efficiency of the autophagy-mediated cellular response to HSV infection. Their results suggest that Us11 competes for binding to the ARF domain with TBK1, thus, inhibiting the efficient binding of TBK1. In an earlier article by the same group, It was reported that Us11 can also target the TBK1 by binding to heat shock protein 90 (hsp90), thereby preventing the formation of the TBK1-hsp90 complex, which is also required for the efficient functioning of the autophagy response mediated by TBK1 [56]. Us11 also binds to protein kinase R [PKR] and together with ICP34.5, which binds to Beclin1 modulate autophagy via the eukaryotic translation initiation factor 2-α kinase (EIF2AK2/PKR) pathway [57]. The dissimilarity between autophagic stimulation of HSV-1 and HSV-2 has been investigated [58]. While HSV-1 inhibits total autophagic activity, HSV-2 maintains basal autophagic activity. Disruption of basal autophagic activity leads to neurodegeneration, which suggests the link between HSV-1 and neurodegeneration. Basal autophagy suggests that autophagy contributes to continuous infection by limiting inflammation, IFN-1 production and NF-_k_β regulation. Subsequently, the persistent latency of HSV-1 in the neuron is maintained by the stimulation of autophagy [58,59].

### 2.2. Interplay of HSV-1 and Host PML Protein

Promyelocytic leukaemia nuclear bodies (PML-NBs), also referred to as nuclear domains 10 (ND10), or PML oncogenic domains (PODs), are small (0.1–1.0 µm) dynamic nuclear structures made of several constant (PML, SP100, Daxx) or transiently associated proteins, depending on the cell function and/or exposed stress [60,61,62]. They respond to varieties of stimuli including apoptosis, senescence viral infections and interferon response [63]. PML-NBs are suggested to be the site of nuclear activities, nuclear protein depots and hotspots for posttranslational modifications [64]. They restrict HSV-1 gene expression and replication by forming structures called viral DNA-containing PML-NBs (vDCP NBs), which is an intrinsic antiviral response strategy of preventing lytic infection [65]. Latency could be attributed to HSV-1 chromatinization mediated by the H3.3 chaperone complex (DAXX/ATRX) [65] and HIRA/UBN1/CABIN1 [66]. It was suggested that the α- thalassemia mental retardation X-linked protein (ATRX) is dispensable in the initial loading of heterochromatin, and is only required for stability maintenance of the viral heterochromatin [67]. Histone cell cycle regulator (HIRA) is suggested to play an independent role in the regulation of intrinsic and innate immune regulation of HSV-1 [68]. Therefore, understanding the underlying mechanisms of HSV-1 chromatinization-induced latency is important. The PML-NBs is organized by protein–protein interaction between SUMOylated proteins and SUMO-interacting motifs (SIMs) [61]. PML is the key protein responsible for the assembly and maintenance of PML-NBs. PML upon SUMOylation recruits other PML-NB-associated proteins to the nuclear structure [62,69,70]. PML and SP100 is activated upon induction by interferon (IFN), leading to an increase in PML-NBs [60]. PML, together with SP100, the death domain-associated protein 6 (DAXX), and ATRX, contributes to the repression of viral gene expression in a cooperative manner [64]. However, this mechanism to repress HSV-1 gene expression is counteracted by ICP0 through its E3 ubiquitin ligase activity. Using a combination of SUMO-dependent and independent targeting strategies, HSV-1 (ICP0) ubiquitinates and degrades PML and SP100, resulting in the disruption of PML-NBs and the subsequent release of viral genes [61].

It is suggested that the biological activity of ICP0 is connected to the host cell SUMOylation events, as mutations of some SIM-like sequences (SLSs) in the c-terminal quarter of ICP0 on HSV-1 reduced ICP0-mediated degradation of PML [71]. The association between ICP0 and PML-NBs is said to be a series of process involving adhesion, fusion and retention [72]. Moreover, ICP0 utilizes different motifs of PML-NBs fusion and SUMO interaction [72], suggesting different molecular mechanisms of PML- ICP0 interaction and degradation. More so, among the PML isoforms, different methods of interactions and degradation were observed to be utilized by ICP0 in Hep-2 and U2OS cells [73]. Hembram and colleagues identified a bona fide SIM in ICP0 which is essential to target SUMOylated PML. It was reported that the phosphorylation of this ICP0 motif by host kinase Chk2 increases the potency of ICP0 to act as a SUMO targeted ubiquitin ligase [STUbl]. Their findings indicate that three post-translational modifications, namely, ubiquitination, SUMOylation and phosphorylation, were involved in an unprecedented crosstalk in the ICP0′s degradation of PML [74]. In another study, Fada and colleagues reported that a PML II mutant lacking both lysine SUMOylation and SIM rendered it unrecognizable by ICP0, preventing ICP0-mediated degradation while still maintaining PML II localization to ND10 [75]. Moreover, it was observed that SP100 degradation was delayed in PML-/-infected cells and that the accumulation of ICP0 was reduced at low but not high multiplication of infection [76]. On the contrary, higher wild-type virus yields were observed in SP100-/-infected cells than in the parental Hep-2 cells at low multiplication of infection [60]. This suggests that HSV-1 may have hijacked the PML stress response function for successful infection. The function of SP100 may be different and could have an important role in suppressing HSV-1 infection. A recent review by Full and Ensser also addressed this interplay between HSV-1 and host PML proteins [77].

### 2.3. Modulation of Apoptosis

Apoptosis is a crucial cellular defence mechanism that ensures the elimination of pathogen-infected cells and has been reviewed by He and Han [78], but there are recent interesting mechanisms of apoptosis modulation by HSV not captured in the aforementioned review that will be outlined in this section. One of the recent findings was on a study that focused on the Herpes simplex encephalitis and the impact of the blood–brain barrier in modulating the pathological development of the disease [79]. It was reported that HSV-1 infection led to the activation of apoptosis with accompanying golgi fragmentation and downregulation of occludin and claudin 5. These cellular responses were mediated by the protein GM130 which was downregulated due to HSV-1 infection [79]. Overexpression of the GM130 attenuated apoptosis, and in cells infected with HSV-1, the protein levels of both occludin and claudin were partially restored. Despite the potentiation of apoptosis reported above, it is established that inhibition of apoptosis is critical for the reactivation of latency by HSV [80], and this mechanism is mediated by the action of the latency-associated transcript (LAT), which is the only HSV transcript expressed significantly during latency. Another recent study showed that CD80 can compensate for the functions of the LAT, such as the establishment of latency, reactivation and immune exhaustion in cells infected with the LAT-null virus [81]. This is significant because this is the first time that such overlapping functionalities between LAT and CD80 have been reported. Although CD80 rescues the functions of LAT, it has distinct functions as CD80 exacerbated eye disease in mice compared to wild-type HSV. Furthermore, CD80 potentiated the expression of the anti-apoptotic Bcl-2 gene thus modulating apoptosis in the infected cells [81]. Another HSV-1 protein that has implications in regulating cellular apoptosis—ICP22 has been shown to function via a mechanism like that used by the cellular J-protein/HSP40 family chaperone [82]. Other HSV proteins that have been implicated in evasion of cellular immune response by interference of apoptosis include Us3 and Us5 [83]. Both proteins interact with critical proteins of the apoptotic pathway. Us3, which is a serine/threonine kinase, inhibits Bad-induced apoptosis, while Us5 is a glycosylated J protein that inhibits Fas-mediated apoptosis [50].

### 2.4. Intracellular Cell-to-Cell Propagation

It has been shown that HSV can make use of extracellular vesicles, not just in mediating intercellular cell to cell propagation, but also in evading the immune system [84]. The manipulation of the MHC class II processing pathway through alteration of the endosomal sorting and trafficking of HLA-DR by HSV-1 has been consistently shown to be an immune evasion mechanism. HSV-1 also exploits the anterograde transport mechanism [84] in cells to move viral particles from neuron cell bodies to axon tips during reactivation of infection, and this is implicated in the spread of viral particles in epithelial tissues and general dissemination of the virus to other hosts. It has been shown that kinesin-1 proteins KIF5A, -5B, -5C play crucial roles in this anterograde transport mechanism. Targeting the transport of reactivating HSV has crucial therapeutic significance [48].

### 2.5. Inactivation in Expression of Signaling Pathways

It has been suggested that HSV-induced Dok phosphorylation and Dok-2 degradation could be a strategy of immune evasion to inactivate T-cells, which might play a role in HSV pathogenesis [85]. Ectopic expression of VP22 was said to decrease cGAS/STING-mediated IFN-β promoter activation and expression of IFN-β [86]. It has been suggested through further studies that VP22 interacts with cGAS and inhibit its enzymatic activity. The γ_1_34.5 protein of HSV-1 was said to inactivate STING and, more so, disrupts its translocation from endoplasmic reticulum to golgi apparatus, an important process necessary to prime cellular immunity [87]. This leads to downregulation of interferon regulatory factor 3 (IRF3) and IFN responses. Another HSV-1 protein, virion host shutoff endonuclease (UL41), was reported to decrease cGAS/STING-mediated IFN-β promoter activation and expression [88]. This protein was reported to degrade cGAS via its RNase activity. It has equally been observed that HSV-1 neuronal infection triggers activation of Src tyrosine kinase, phosphorylation of dynamin 2 GTPase and perturbation of golgi apparatus (GA) integrity [89]. A scattered and fragmented distribution of the GA through the cytoplasm with swollen cisternae and disorganized stacks was observed in HSV-1-infected neurons in contrast with the uniform perinuclear distribution pattern observed in control cells [89]. The evasion of HSV-1-specific CD8+ T cells which accumulates in infection sites is enhanced by HSV-1 UL13 kinase through reducing the expression of the CD8+ T cell attractant chemokine CXCL9 in the CNS of infected mice, leading to increased HSV-1 mortality due to encephalitis [90]. VP24 protein was reported to dampen interferon stimulatory DNA (ISD)-triggered IFN-β production and inhibit IFN-β promoter activation induced by cyclic cGAS and STING [91]. The ectopic expression of VP24 selectively blocked interferon regulatory factor 3 (IRF3) and downregulated ISD-induced phosphorylation and dimerization of IRF3 during HSV-1 infection in a VP24 stable knockdown human foreskin fibroblast cell line [91]. Here, VP24 disrupts the interaction between TANK- binding kinase 1 (TBK1) and IRF3, impairing IRF3 activation. It has been reported that HSV-1 downregulates CD1d cell surface expression and suppresses the function of NKT cells through its viral protein kinase US3 [92]. US3 phosphorylates KIF3A at serine 687, leading to downregulation of CD1d expression. The interferon-induced protein with tetratricopeptide repeat 3 (IFIT3) is an antiviral host intrinsic factor that restricts replication of DNA and RNA viruses. UL41 (HSV-1 tegument protein) was reported to inhibit the antiviral activity of IFIT3 [93]. UL41 diminishes the accumulation of IFIT3 mRNA to abrogate its antiviral activity. A study reported CD1d expression downregulation and subsequent suppression of NKT cells function, using its viral serine/threonine protein kinase US3 as another strategy of the virus to evade immune response [94]. Peroxisomes are thought to be important signalling platforms of antiviral innate immunity, as signalling from peroxisomal MAVS (MAVS-Pex) triggers a rapid production of IFN-independent and -stimulated genes (ISGs) against invading pathogens. Another study also revealed that HSV-1, through its tegument protein VP16, blocks MAVS-Pex-mediated early ISG expression to dampen the immediate early antiviral innate immunity signalling from peroxisomes [95].

### 2.6. Role of miRNA in HSV Immune Evasion

MicroRNA (miRNA) is a short (20–24 nucleotides) non-coding RNA that is involved in post-transcriptional regulation of gene expression in multicellular organisms. MicroRNA 146a (miR146a) inhibits the expression of STAT1. However, Nuclear Dbf2-related kinase 1 (NDR1), which promotes virus-induced production of type 1 IFN, proinflammatory cytokines and ISGs, enhances STAT1 translation by binding to the intergenic region of miR146a. This leads to inhibition of the expression of miR146a, subsequently liberating STAT1 from miR146a-mediated translational inhibition [96]. More so, STAT1 binds to the promoter of miR146a, decreasing its expression. A study investigated the role of LAT encoded miRNAs in resistance to apoptosis and establishment of latent infection [97]. Five miRNAs (miR-H3, miR-H4-3p, miR-H4-5p, miR-H24, and miR-H19) encoded by latency associated transcript RLI sequence are implicated due to the overexpression of miR-H3, miR-H4-5p and miR-H19 in PC12 cells [97]. Six upregulated miRNAs (miR-592, miR-1245b-5p, miR-150, miR-342-5p, miR-1245b-3p and miR-124) were reported to downregulate TLR pathway-associated genes following HSV-2 infection [98]. Host-encoded miR-649 was reported to promote HSV-1 replication through regulation of the mucosa-associated lymphoid tissue lymphoma translocation gene 1 (MALT1)-mediated antiviral signalling pathway [99]. Micro-RNA-155 (miR-155) was reported to contribute to the pathogenesis of stomal keratitis [100]. HSV-1-encoded miRH8 was reported to target PIGT of the glycosylphosphatidylinositol (GPI), resulting in reduced expression of the antiviral protein tetherin and GPI-anchored activation of NK cell ligands, as well as a subsequent decrease in viral recognition and elimination by NK cells [101], which led to enhanced viral spread. miR-23a binds to the three prime untranslated region (3′ UTR) of interferon regulatory factor 1 (IRF1) to downregulate its expression and, thus, facilitates HSV-1 spread [102]. miR-221 negatively regulates IFNβ production at the time of virus production and miR-221 is induced by ELF4 by binding to the miR-221 promoter [103]. miRNA-H4-5p binds to 3′ UTR of CDKN2A and CDKL2, which reduces their expression and subsequently leads to reduced apoptosis and cell cycle progression [104].

Moreover, miR-H6 targeting of ICP4 inhibits HSV-1 productive infection and a decrease in production of IL-6 in human cornea epithelial (HCE) cells [105]. A group of researchers reported using hsa-miR-7704, expressed on macrophage, to inhibit HSV-1 in infected HeLa cells [105]. miRNA401 delivered to cells through exosomes were demonstrated to reduce viral yields via targeting ICP4 [106], an essential viral regulatory protein in a dose-dependent manner. miR-101, an ICP4-induced expression, was reported to downregulate RNA-binding protein G-rich sequence factor 1 (GRSF1) expression, inhibiting the replication of HSV-1 [107], as binding of GRSF1 to HSV-1 p40 mRNA, enhances its expression and viral proliferation [107].

## 3. HSV Vaccination and Immunotherapies

### 3.1. HSV Vaccination

Vaccination against HSV virus has been a far-fetched success, as there are currently no approved vaccines against the virus due to the reason that those produced have not successfully passed the clinical trials for safe use in humans. The recent interest in HSV vaccine development could be attributed to the recorded success in the vaccine against varicella zoster virus (VZV) and because both viruses (HSV and VZV) are alpha-herpes viruses, sharing similar pathogenesis pattern. Both viruses infect the skin and nerves, develop latent infections in the trigeminal and dorsal root ganglia, and have a tendency to reactivate. Therefore, there seems to be intensified efforts towards HSV vaccine development. Different kinds of vaccines have been developed for the treatment and/or prevention of HSV (Table 1), however none has been licensed for human use.

### 3.2. Potential Vaccine Candidates

There are several recent reviews that addressed the advancement of HSV vaccine design and discussed numerous emerging HSV candidate vaccines [144,145,146]. In this section we provide an update to this discussion by highlighting the new published findings in the search for novel prophylactic and therapeutic HSV vaccines not captured in prior reviews.

Different approaches being employed in candidate vaccine development include subunit vaccines, replication incompetent viruses and live attenuated vaccines [45]. It has been reported that HSV-1 mutant strains carrying modified Us3 and Us5 when used to infect mice triggered an asymptomatic immune response when challenged with the wild-type virus [145]. Interestingly, this immune response led to significantly less inflammatory cell aggregation in nervous tissues compared to the wild-type HSV-1 which induced an intense inflammatory reaction [145].

An exciting new study reports the development of a novel DNA vaccine against HSV-2 [146]. Their DNA vaccine encoded the following viral proteins: gB, gD, ICP0, ICP4 and UL39. It was reported that upon inoculation of the test animals with the vaccine, significant T cell response was generated. CD4^+^ T cells and antigen-specific CD8^+^ were stimulated to produce high amounts of IFN-γ, both in locally infected tissues and lymphoid organs. The authors also report the generation of memory T cells which are being studied to evaluate whether they can mount an anti-HSV-2 immune response even after primary infection [147]. Recently, a group of researchers tested their DNA vaccine candidate—a codon-modified polynucleotide vaccine COR-1 in HSV-2 positive patients, and reported a reduction in viral shedding after administration [115]. The authors had earlier shown that COR-1 induces a balanced adaptive humoral and cell-mediated immune response in mice, and protected mice challenged with a lethal dose of HSV-2 [142], and was also shown to elicit minimal adverse effects when tried in healthy volunteers [116]. This positive outcome of COR-1 in HSV-2 positive patients lends credence to the potential of COR-1 as a HSV therapeutic vaccine. Another group in their study explored the possibility of maternal immunization with the single-cycle HSV candidate vaccine deleted in glycoprotein-D [148,149,150], which induces antibody-dependent cell-mediated cytotoxicity (ADCC) to confer protection to neonates. It has been reported that the vaccine significantly protected new-born mice from neonatal HSV, and showed that the neonatal protection was a consequence of the new-borns acquiring antibodies that mediate ADCC from the mother transplacentally and through breastfeeding [151].

In a clinical trial funded by Genocea, a group of researchers showed that a candidate therapeutic vaccine, GEN-003, which is a purified protein subunit vaccine made by deleting a large fragment of ICP4 in addition to a transmembrane deletion mutant of gD [152], generated promising outcomes. It was reported that the candidate vaccine stimulated both humoral and cellular immune responses while exhibiting minimal adverse effects in the HSV-2 seropositive persons with genital herpes [4]. Another candidate vaccine—an intranasal vaccine comprising of HSV-2 surface glycoproteins gD2 and gB3, has also been investigated. The vaccine was formulated in a nanoemulsion adjuvant. Using HSV-2 genital herpes in a guinea pig model, the authors reported positive outcomes for their candidate vaccine. There was a higher level of neutralizing antibodies compared to the single-surface glycoprotein candidate vaccine when both were injected into the guinea pig model. Furthermore, there was a significant reduction in the ability of the challenge of HSV-2 establishing latent infection in the dorsal root ganglia of the vaccinated guinea pigs [153].

Another promising candidate vaccine was also reported by the same group of researchers. In this study using a guinea pig HSV genital model, it was demonstrated that the candidate vaccine, which contained highly purified, inactivated HSV-2 particles (with and without additional recombinant glycoprotein D), provided protection against different HSV-2 homologous and heterologous strains. Importantly, it was reported that upon challenge, the candidate vaccine conferred protection against virus retention in the ganglia and spinal cords of most animals. The results also suggest that there was no added advantage of the glycoprotein D on the efficacy of the candidate vaccine [154].

### 3.3. Possible Immunotherapies

Understanding the purpose for which immunotherapies are developed for HSV infection is a key step to successful infection prevention and control. On the premise that prophylactic and immunotherapeutic vaccines have different goals and initiate different immune responses (carefully discussed by Truong et al. [144]), it was suggested that therapeutic vaccines have a lesser immunological task compared to the prophylactic vaccine. On the contrary, another group of researchers [8] argued that therapeutic vaccines are often faced with multiple immune evasion mechanisms by successfully establishing HSV infection to persist in host cells. The two groups have different views on the level of immunological tasks of prophylactic and immunotherapeutic vaccines and on the research efforts that should be focused on. Based on this contrasting opinion and on the mechanism of establishment of infection by HSV carefully discussed in this article, it is important to focus on prevention and treatment based on the presented pattern of infection of HSV over the years. More so, preventing the acquisition and colonization of the dorsal root ganglia could prevent the event of virus latency and subsequent reactivation. It is necessary to consider that HSV-1 and HSV-2 show different patterns of infection and immune invasion, as well as that the establishment of latency and viral shedding are exploited in different ways by these viruses at different sites. Therefore, it is of paramount importance to understand the type of HSV infection and the site of action, the type of effector immune response, and to identify the immunogenic pathogen proteins, the mechanism of immune evasion and the signalling pathways involved, the needed DCs and/or T cell to be targeted to stimulate response, how to target and activate the DCs, how and where the designed adjuvants will work and the possibility of off-targets which may lead to cytotoxicity. Furthermore, designing a vaccine that is both therapeutic and protective is of paramount importance for virus control and prevention. Prophylactic vaccines for HSV are needed to effectively stimulate primary immune responses at the site of pathogen entry, involving the naïve T cells of which the DCs are the important cell type for stimulating naïve T and B cells [144], thereby protecting uninfected hosts. On the other hand, immunotherapeutic vaccines are also important to reduce herpes shedding and alleviate herpetic disease in symptomatic patients with recurrent outbreaks [8]. Considering this and the viral survival mechanisms for the establishment of infection as already discussed, it is pertinent to design immunotherapies to solve both treatment and prevention problems.

## 4. Conclusions

Virus invasion is normally followed by activation of both the innate and adaptive immune systems, which through the production of NK cells, recognize the glycoprotein present on HSV. Through this and in the presence of TLR and other important cells, IFN-γ as well as numerous other important immune cells (as outlined in different parts of this review) are produced. On the other hand, HSV uses different mechanisms, including inhibition of induction of autophagy and apoptosis to avoid the immune system and maintain itself in latency. Designing both therapeutic and protective vaccine is of paramount importance for virus control and prevention. Prophylactic vaccines for HSV are needed to effectively stimulate primary immune responses at the site of pathogen entry. Among the naïve T cells, the DCs are the important cell type for stimulating naïve T and B cells [144], and this stimulation subsequently leads to the protection of uninfected hosts. On the other hand, immunotherapeutic vaccines are also important for the reduction in herpes shedding and alleviating herpetic disease in symptomatic patients with recurrent outbreaks [8]. The existing challenges related to the deployment of a HSV vaccine in humans include the expensive costs of carrying out trials to test the efficacy of HSV vaccine candidates, and this is especially critical, as most of the candidates that exhibited potency in animal models did not confer same protection in humans [155]. Another challenge is the uncertainty of the humoral and cell-mediated responses induced by HSV vaccine candidates to confer lasting protection. Furthermore, the development of vaccine candidates that can confer protection against both HSV-1 and HSV-2, for example in the case of genital herpes, remains a challenge. Different promising vaccines are already undergoing testing or have completed testing, and it is hoped that this review has contributed to the understanding of the interplay between the immune response and the virus evasion mechanism, which is necessary for the development of an efficient vaccine against HSV.

## Figures and Tables

**Figure 1 vaccines-08-00302-f001:**
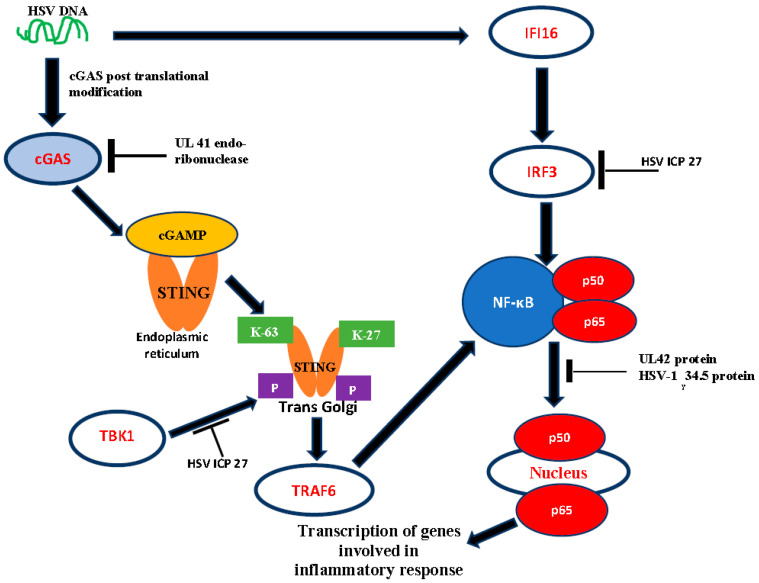
Cytosolic DNA sensing and pathway activation. HSV UL41 endoribonuclease breaks down. cGAS mRNA inhibits it by reaching a threshold required to activate cGAMP production. HSV ICP 27 protein inhibits TBK1 kinase by phosphorylating oligomerized STING, meaning that TRAF6 will not be recruited, subsequently inhibiting NF-κB activation. Additionally, HSV ICP 27 inhibits IRF3 in an alternative pathway that activates NF-κB. UL42 and HSV-1 γ34.5 protein inhibits the translocation of p50 and p65 subunits of NF-κB by inhibiting the transcription of genes involved in inflammatory response. UL41: Virion host shutoff (Vhs); cGAMP: Cyclic guanosine monophosphate (GMP), Adenosine monophosphate (AMP); cGAS mRNA: Cyclic GAMP synthase messenger ribonucleic acid; ICP: Infected cell protein; TBK1: TANK-binding kinase 1; STING: Stimulator of interferon genes; NF-kB: Nuclear factor kappa B; IRF: Interferon regulatory factor 3.

**Table 1 vaccines-08-00302-t001:** Types of vaccines that have been developed against herpes simplex virus.

S/N	Name of Vaccine	Type of Vaccine	Antigens	Adjuvants	Mode of Action	Phase of Trial	Company/Institute
1	GEN-003 [4,108,109]	Therapeutic	gD2, ICP4	Matrix-M2	Stimulates both humoral and cellular immune response	Phase II	Genocea Biosciences
2	HSV529(ACAM 529)/d15-29 [7,110,111,112,113,114]	Prophylactic	Replication-deficient derived from dl5-29	Not Applicable	Stimulates production of neutralizing antibody and mild CD4+ T-cells	Phase I	Sanofi
3	COR-1 [115,116]	Therapeutic	HSV-2 DNA	Vaxfectin	Cell-mediated immune response	Phase I/IIa	VGXI Inc. (Texas, USA) under license from Admedus
4	Trivalent Vaccine [5,117,118]	Therapeutic	gC2, gD2, gE2	CpG and alum	Blocks virus entry by gD2 and immune evasion by gC2 and gE2. Induces plasma- and mucosa-neutralizing antibodies, stimulates CD4 T cell response	Clinical phase	Harvey M. Friedman Penn Institute of Immunology, University Pennsylvania
5	gD2 subunit vaccine [6,119,120,121,122,123]	Prophylactic	gD2	AS04, MPL and alum,	Produces neutralizing antibodies to gD2	Phase III	Glaxosmithkline
6	KOS-NA [124,125]	Prophylactic	Mutation in *UL39* encoding ICP6 (Live attenuated)	Not applicable	Anti-apoptosis effect as a result of diminished ICP6 protein levels	Pre-clinical	David J. David (University of Kansas, USA) and Lynda Annemarison (St. Louis University, USA)
7	HSV-2 CJ2-gD2 [126,127]	Prophylactic	Replication defective, expressing gD2	Not applicable	Elicits neutralizing antibody	Pre-clinical	Department of Surgery and the Department of Medicine Brigham Hospital and Women Hospital and Harvard Medical School, Boston
8	HerpV [128,129]	Therapeutic	32 HSV-2 peptides	QS-21	Elicits CD4^+^ and CD8^+^ T cell responses	Completed Phase II	Agenus
9	G103 [130]	Prophylactic	gD, UL19, UL25	GLA	Elicits antigen-specific binding and neutralizing antibody responses, including CD4 and CD8 effector and memory T cells	Pre-clinical	Immune-Design (Sanofi)
10	VC2 [131,132,133]	Prophylactic	Mutations in gK and UL20 (life attenuated)	Not Applicable	Induces humoral and cellular immunity	Pre-clinical	Lousiana State University
11	Vaxfectin^®^- gD2/UL46/UL47 [134,135]	Prophylactic	gD, VP11/12, VP13/14	Vaxfectin	Induces neutralizing antibody and stimulates CD8+ T cells	Phase II	Vical
12	RR2 [8]	Therapeutic	RR2 protein	CPG and alum	Boosts high neutralizing antibodies, enhance number of functioning IFN-γ	Pre-clinical	-
13	HSV-2 0∆NLS [136,137]	Prophylactic	Live HSV-2 ICP0^-^ (Live attenuated)	Not applicable	Stimulates the humoral and cellular immune response	Phase I	Rational Vaccines Inc. (RVs)
14	sgG-2 [138,139]	Prophylactic vaccine candidate	gD	CpG and alum	Stimulates IgG antibody response	Pre-clinical	-
15	gE2 [140]	Prophylactic	gE	Live attenuated gE deletion mutant	CPG and alum	Stimulates neutralizing antibody	-
16	gB1s-NISV [141]	Therapeutic	gB	CpG	Generates gB-specific IgG antibody and lymphoproliferative responses	Pre-clinical	-
17	Codon optimized polynucleotide vaccine [142]	Therapeutic	gD	Plasmid encoded	Induces both B and T cell responses	Phase II	Admedus
18	VR∆41 [143]	Prophylactic	Live attenuated	Not Applicable	Spreads to the CNS from the site of inoculation, evoke potent immune reaction within the CNS without the induction of lethal encephalitis	Pre-clinical	Fukushima Medical University School of Medicine, Japan

Keys: GEN-003: HSV protein subunit vaccine consisting of 2 recombinant T cell antigens: ICP4 and gD; ICP4: Infected cell protein 4; ACAM 529: HSV-2 replication-defective vaccine with UL5 and UL29 deleted; CD: Cluster of differentiation; COR: Codon-modified and optimized plasmid; gD2: Glycoprotein D 2; MPL: Monophosphoryl lipid A; KOS-NA: Mutant HSV-1 containing novel mutations in the UL39 gene; CJ2-gD2: A novel non-replicating dominant-negative HSV-2 recombinant viral vaccine; QS-21: Active fraction of the bark of Chilean tree, Quillaja saponaria; G103: HSV-2 vaccine that consists of 3 recombinantly expressed HSV-2 proteins (gD, UL19 and UL25 gene products; GLA: Glucopyranosyl lipid adjuvant; VC-2: HSV-1 live attenuated vaccine; CPG: short single stranded DNA molecules that contain cytosins triphosphate and guanine triphosphate with a phosphodiester link between them; HSV-2 0ΔNLS: HSV-2 ICP0 negative mutant; SgG: HSV-2 cleaved to a secreted amino-terminal portion; gE2: Glycoprotein E 2; gB1s-NISV: Intranasal non-ionic surfactant vesicles containing recombinant HSV-1 glycoprotein B; VRΔ41: UL41-deleted recombinant HSV-1 strain; CNS: Central nervous system.

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
