# Peer review of "Immune Response to Herpes Simplex Virus Infection and Vaccine Development"

_vaccines, 2020, doi:10.3390/vaccines8020302_

Round 1
Reviewer 1 Report
Ike and colleagues have reviewed recent advances in interaction between human herpesvirus 1 and the immune system. While the paper is well written and there are very useful information in the paper, I thought to bring below information to the attention of the authors.
My main comment is that the manuscript is not cohesive. the sections are not very nicely integrated into the paper. Also, I would like to see a new section on the interplay between HHV-1 and host PML proteins. The authors are mentioning the interaction between HHV-1 and the host to control autophagy, but there is no mention of PML.
Some minor points:
Lack of line numbers make it difficult to refer to specific lines.
In various places, the authors refer to the author of other papers as "they have reported..". Please consider to use "It is reported" instead or Name Author and colleagues reported.
Page 3, VHS, not Vhs.
Figures are not consistent. Figure 1 is used twice in pages 3 and 4.
Reference styles are not consistent and needs to be redone. Consider to use a reference management software.
Below sentence in section 5, last paragraph: “… an efficient vaccine against HSV should not be far from approval. “ is speculative. Please consider to remove it.
Author Response
Reviewer’s Comment: My main comment is that the manuscript is not cohesive. the sections are not very nicely integrated into the paper. Also, I would like to see a new section on the interplay between HHV-1 and host PML proteins. The authors are mentioning the interaction between HHV-1 and the host to control autophagy, but there is no mention of PML.
Author’s Answer: We have worked to integrate the different sections into the paper by deleting some sections and integrate some parts into others. A new section on PML, Section 2.2: Interplay of HSV-1 and host PML has been included.
Reviewer’s Comment: Lack of line numbers make it difficult to refer to specific lines.
Author’s Answer: Line numbers have been included for all the pages.
Reviewer’s Comment: In various places, the authors refer to the author of other papers as "they have reported..". Please consider to use "It is reported" instead or Name Author and colleagues reported.
Author’s Answer: All the places were “they have reported..” were used have been modified according to reviewer’s recommendations. Examples: Page 4 Line 30: “They reported..” has been changed to “It has been reported..; Page 12 Line 27 same as above.
Reviewer’s Comment: Page 3, VHS, not Vhs.
Author’s Answer: Vhs has been changed to VHS, Page 3 Line 30.
Reviewer’s Comment: Figures are not consistent. Figure 1 is used twice in pages 3 and 4.
Author’s Answer: All the authors have agreed to remove the second figure in page 4. It was not referred to in text and was irrelevant.
Reviewer’s Comment: Reference styles are not consistent and needs to be redone. Consider to use a reference management software.
Author’s Answer: References have been redone and are now consistent.
Reviewer’s Comment: Below sentence in section 5, last paragraph: “… an efficient vaccine against HSV should not be far from approval. “ is speculative. Please consider to remove it.
Author’s Answer: This sentence has been modified and the speculation removes. Page 13 Lines 48-51.
Reviewer 2 Report
Vaccine development against viruses is a topic that people are caring about now. HSV vaccine development remains a major challenge. Understanding the complex interactions between the immune response and HSV can help vaccine development. The authors summarized the recent progress in the field of the immune response to Herpes Simplex Virus infection and vaccine development in this manuscript. Generally, the authors are providing a well-written review manuscript. They have included the most recent and critical references related to the topic. However, there are several concerns that need to be addressed. Some parts are relatively redundant, for example, the cGAS-Sting pathway is repeatedly discussed in many sections. The manuscript also lacks a part discussing the responses of immune cell populations to HSV-1 infection.
- It will be a great help if the authors could add line number to text in the manuscript so that the readers can easily point out the exact position.
- It is confusing that there are 2 Figure 1 and I did not find the information or legend for the later one. Please correct it.
- Part of “1.3 The interplay of herpes simplex infection and the immune system”. ICP0 is an essential HSV-1 antigen that antagonizes host Immune signaling pathways and intrinsic host defenses (IFI16, MyD88, NF-κB, PML body, etc) through its ubiquitin ligase functions. There were some cases that researchers tried to use ICP0 mutant HSV-1 as vaccines (Derek J. Royer et al, 2016; William P. Halford et al, 2011). It would be better if the authors could demonstrate more about ICP0’s roles in HSV-1 infection and vaccine development.
- In the part of “1.3 The interplay of herpes simplex infection and the immune system”, the authors have focused on the interactions between HSV-1 and intracellular immunity. Specifically, they focused on the review of how HSV-1 regulates TNF-a pathway among the inflammatory response. What about other inflammatory cytokine responses? And, it would be a great help if they can provide some reviews on the interactions between HSV-1 infection and different immune cells, especially the immune response of T cells during HSV-1 infection. In addition, what are the possible roles of the immune system in the HSV-1 reactivation?
- HSV-1 can suppress host antiviral mechanisms, replicate in epithelial cells and propagate cell-to-cell. HSV-1 establishes latency infection in the neurons of both the peripheral and central nervous systems. During latency, HSV-1 only highly expresses a non-coding RNA (LAT). LAT can not trigger potent immune responses although it may play a role in modifying CD8+ T cell effector activity.The property of HSV-1 establishing latency infection without triggering immune response might block the vaccine development, how to overcome the potential obstacles during vaccine development?
- In the part of “2.1. Innate and adaptive immune system to HSV infection”, the review lacks the response of the adaptive immune system to HSV infection. Please include this part.
- In the part of “2.1. Innate and adaptive immune system to HSV infection”, plasmacytoid dendritic cells (pDGs) should be “pDCs”.
- The part of “2.2 HSV signaling and regulatory pathways” did not provide substantial information and it seems like an unnecessary section.
- The authors may need to combine the section “2.5 Apoptosis” and “3.2. Modulation of apoptosis”. The authors did not demonstrate the relationship between HSV-1 induced cellular survival and immune responses in the section of “2.5 Apoptosis”. People might not understand why the authors include “2.5 Apoptosis” in the section of “Immune response to HSV infection”. The authors showed the relationship in the latter section, but it is kind of confusing to readers.
- The Ref. 98 mainly demonstrates the role of miR-H4b in regulating cell proliferation and cell cycle by targeting p16 mRNA. It is not an appropriate reference in the section of “Role of miRNA in HSV immune evasion” .
- It would be helpful for the readers if the authors can include the names of institutes or companies that are developing the vaccines listed in Table 1.
Author Response
Reviewer’s Comment 1: It will be a great help if the authors could add line number to text in the manuscript so that the readers can easily point out the exact position.
Author’s Answer: Line numbers have been included for all the pages.
Reviewer’s Comment 2: It is confusing that there are 2 Figure 1 and I did not find the information or legend for the later one. Please correct it.
Author’s Answer: It has been corrected. All the authors have agreed to remove the second figure in page 4. It was not referred to in text and was irrelevant.
Reviewer’s Comment 3: Part of “1.3 The interplay of herpes simplex infection and the immune system”. ICP0 is an essential HSV-1 antigen that antagonizes host Immune signaling pathways and intrinsic host defenses (IFI16, MyD88, NF-κB, PML body, etc) through its ubiquitin ligase functions. There were some cases that researchers tried to use ICP0 mutant HSV-1 as vaccines (Derek J. Royer et al, 2016; William P. Halford et al, 2011). It would be better if the authors could demonstrate more about ICP0’s roles in HSV-1 infection and vaccine development.
Author’s Answer: The role of ICP0 in viral replication and reactivation of latency has been added to section 1.3, Page 3 Lines 14-20. Additional functions of ICP0 have also been added in a new section on the interplay of HSV-1 and host PML protein, Page 6 Lines 34-53.
Reviewer’s Comment 4: In the part of “1.3 The interplay of herpes simplex infection and the immune system”, the authors have focused on the interactions between HSV-1 and intracellular immunity. Specifically, they focused on the review of how HSV-1 regulates TNF-a pathway among the inflammatory response. What about other inflammatory cytokine responses? And, it would be a great help if they can provide some reviews on the interactions between HSV-1 infection and different immune cells, especially the immune response of T cells during HSV-1 infection. In addition, what are the possible roles of the immune system in the HSV-1 reactivation?
Author’s Answer: The roles of other inflammatory cytokines responses and the interaction between HSV-1 and other immune cells, especially immune response of T cells have been added. Page 3 Lines 36-51 and Page 4 Lines1-2.
Reviewer’s Comment 5: HSV-1 can suppress host antiviral mechanisms, replicate in epithelial cells and propagate cell-to-cell. HSV-1 establishes latency infection in the neurons of both the peripheral and central nervous systems. During latency, HSV-1 only highly expresses a non-coding RNA (LAT). LAT can not trigger potent immune responses although it may play a role in modifying CD8+ T cell effector activity. The property of HSV-1 establishing latency infection without triggering immune response might block the vaccine development, how to overcome the potential obstacles during vaccine development?
Author’s Answer: A section on the role of ICP0 in latency reactivation was added. The role of LAT in reactivation of latency was also discussed in Section 2.3, Page 7 lines 13-21 but it did not directly address how latency infection without immune response can be overcome. Authors could not get any published report that addressed this issue.
Reviewer’s Comment 6: In the part of “2.1. Innate and adaptive immune system to HSV infection”, the review lacks the response of the adaptive immune system to HSV infection. Please include this part.
Author’s Answer: The response of the adaptive immune system to HSV has been added. Page 3 Lines 43-51
Reviewer’s Comment 7: Innate and adaptive immune system to HSV infection”, plasmacytoid dendritic cells (pDGs) should be “pDCs”.
Author’s Answer: This section has been integrated into 1.3 and the words removed. pDGs is no more in the text.
Reviewer’s Comment 8: The part of “2.2 HSV signaling and regulatory pathways” did not provide substantial information and it seems like an unnecessary section.
Author’s Answer: Authors agree, this section has been removed.
Reviewer’s Comments 9: The authors may need to combine the section “2.5 Apoptosis” and “3.2. Modulation of apoptosis”. The authors did not demonstrate the relationship between HSV-1 induced cellular survival and immune responses in the section of “2.5 Apoptosis”. People might not understand why the authors include “2.5 Apoptosis” in the section of “Immune response to HSV infection”. The authors showed the relationship in the latter section, but it is kind of confusing to readers.
Author’s Answer: The two sections have been combined, now section 2.3.
Reviewer’s Comment 10: The Ref. 98 mainly demonstrates the role of miR-H4b in regulating cell proliferation and cell cycle by targeting p16 mRNA. It is not an appropriate reference in the section of “Role of miRNA in HSV immune evasion”.
Author’s Answer: The sentences and the reference have been removed.
Reviewer’s Comment 11: It would be helpful for the readers if the authors can include the names of institutes or companies that are developing the vaccines listed in Table 1.
Author’s Answer: Names of institutes or companies that are developing the vaccines have been provided for most of the vaccines. Only for four of the vaccines could this not be found, Table 1.
Round 2
Reviewer 2 Report
I have carefully re-read the revised manuscript and point-for-point, and the authors have adequately addressed my major concerns. They now provide additional discussions on ICP0’s roles in HSV-1 infection and the responses of T cells to HSV-1. They also removed or combined the paragraphs which were raised by the reviewers. The efforts in reorganizing the manuscript structure have largely strengthened the paper.
This manuscript is a resubmission of an earlier submission. The following is a list of the peer review reports and author responses from that submission.